# Effects of Stepwise Temperature Shifts in Anaerobic Digestion for Treating Municipal Wastewater Sludge: A Genomic Study

**DOI:** 10.3390/ijerph19095728

**Published:** 2022-05-08

**Authors:** Gede Adi Wiguna Sudiartha, Tsuyoshi Imai, Yung-Tse Hung

**Affiliations:** 1Graduate School of Sciences and Technology for Innovation, Yamaguchi University, Yamaguchi 755-8611, Japan; adiwiguna.sudiartha@unud.ac.id; 2Environmental Engineering Study Program, Faculty of Engineering, Udayana University, Bali 80361, Indonesia; 3Department of Civil and Environmental Engineering, Cleveland State University, FH 112, 2121 Euclid Ave, Cleveland, OH 44115, USA; y.hung@csuohio.edu

**Keywords:** anaerobic digestion, biogas production, genomic analysis, shifted-up temperature, sludge treatment and disposal, thermotolerant bacteria

## Abstract

In wastewater treatment plants (WWTP), anaerobic digester (AD) units are commonly operated under mesophilic and thermophilic conditions. In some cases, during the dry season, maintaining a stable temperature in the digester requires additional power to operate a conditioning system. Without proper conditioning systems, methanogens are vulnerable to temperature shifts. This study investigated the effects of temperature shifts on CH_4_ gas production and microbial diversity during anaerobic digestion of anaerobic sewage sludge using a metagenomic approach. The research was conducted in lab-scale AD under stepwise upshifted temperature from 42 to 48 °C. The results showed that significant methanogen population reduction during the temperature shift affected the CH_4_ production. With 70 days of incubation each, CH_4_ production decreased from 4.55 L·g^−1^-chemical oxygen demand (COD) at 42 °C with methanogen/total population (M·TP^−1^) ratio of 0.041 to 1.52 L·g^−1^ COD (M·TP^−1^ ratio 0.027) and then to 0.94 L·g^−1^ COD ( M·TP^−1^ ratio 0.026) after the temperature was shifted to 45 °C and 48 °C, respectively. *Methanosaeta* was the most prevalent methanogen during the thermal change. This finding suggests that the *Methanosaeta* genus was a thermotolerant archaea. *Anaerobaculum*, *Fervidobacterium,* and *Tepidanaerobacter* were bacterial genera and grew well in shifted-up temperatures, implying heat-resistant characteristics.

## 1. Introduction

Water resources and environmental protection policies worldwide have mandated thorough treatment of wastewater prior to discharge into water bodies [1]. Activated sludge treatment is a common wastewater treatment method [2]. However, the main issue regarding this type of treatment is sludge generated from primary sedimentation (PS) and activated sludge (AS). Sludge produced from wastewater treatment plants (WWTP) is produced in large volumes worldwide with up to 8910, 6510, 2960, 650, 580, 550, and 370 thousand metric tons of dry sludge produced annually by EU countries, the United States, China, Iran, Turkey, Canada, and Brazil, respectively [3]. Considering the substantial volumes of waste production, it is not surprising that WWTP sludge management and disposal have become an area of significant concern globally [4]. As a result of its high water content, low dewaterability, and rigorous regulations for sludge reuse and disposal, sludge management is a demanding and complicated issue in wastewater treatment plants [5].

In some countries, landfill is the most favorable disposal method [6]. However, due to the volume of sludge produced and the availability of the land area, attention has shifted to the development of other potential usable products. Currently, the wastewater treatment paradigm has shifted to an environmentally friendly process to reduce the volume of sludge disposed and convert it into a bioenergy source. The dry bulk of WWTP sludge contains organic components that can be utilized to generate a significant quantity of biomass energy [7]. Anaerobic digestion (AD) is one of the most reliable and promising technologies [8], with several biological wastewater treatment plants applying it as an end-treatment for sewage sludge, primary sludge, and waste-activated sludge [9,10,11]. In Japan, especially in Ube wastewater treatment plants, sludge generated in PS and AS is dewatered and delivered to an anaerobic digester.

AD has several benefits compared to other biological processes, such as effortless operation, potentially creating an organic by-product that may be utilized in agriculture and managed to provide an appropriate treatment process [12,13]. During anaerobic digestion, organic compounds are hydrolyzed into soluble fermentable substrates, which are subsequently fermented to acetate, carbon dioxide (CO_2_), and hydrogen gas (H_2_) by acetogenic and acidogenic bacteria. These products are then consumed by methanogens to generate methane (CH_4_) [9]. Silva et al. [14] investigated the CH_4_ and biohydrogen production from a mixture of food waste, anaerobic sewage sludge, and glycerol. The maximum yield of CH_4_ and biohydrogen obtained from the mixture was 342.0 mL CH_4_·g^−1^ vs. and 179.3 mL H_2_·g^−1^ VS, respectively. Without any substrate addition, anaerobic sludge was also capable of generating biogas at 230 ± 29 mL·L^−1^·d^−1^ with CH_4_ production of 153 mL·L^−1^·d^−1^ [15].

There are several temperature conditions where anaerobic digestion frequently occurs at psychrophilic (<30 °C), mesophilic (30–40 °C), and thermophilic (50–60 °C) [16]. Recent studies have attempted to investigate the potential of biogas production from anaerobic sludge at various temperatures. Mirmasoumi et al. [17] investigated the biogas production of anaerobic sludge under two different conditions: mesophilic condition (37 °C) and thermophilic condition (55 °C). Under mesophilic conditions, the maximum CH_4_ produced was 0.246 m^3^ CH_4_·m^−3^ per digester per day. Meanwhile, a greater CH_4_ productivity was obtained under thermophilic conditions, up to 0.64 m^3^ CH_4_/m^3^ per digester per day. Kasinski [18] also found higher CH_4_ yields under thermophilic conditions than under mesophilic conditions (0.56 L CH_4_·g^−1^ vs. −0.70 L CH_4_·g^−1^ VS and 0.25 L CH_4_·g^−1^ vs. −0.32 L CH_4_·g^−1^ vs., respectively). These findings suggest that biogas production using anaerobic digestion generally favors high temperatures. However, in a large-scale WWTP, maintaining thermophilic conditions in the reactor during anaerobic digestion will require significant energy, which will lead to higher operational expenses, especially in four-seasoned countries. In some cases, fermentation failure can occur owing to transient temperature increases caused by power outages, mechanical faults, or human errors during the fermentation process [19]. This motivates further extensive studies of thermotolerant microorganisms for the anaerobic digestion process.

Thermotolerant microorganisms are microbial consortia that are robustly adapted to harsh conditions during industrial applications [20]. Thermotolerant microorganisms are mostly mesophilic, with optimum growth temperatures of 35–45 °C, which are 5–10 °C higher than typical mesophilic strains of the same genus [21,22,23,24]. These strains cannot be classified as thermophilic microorganisms, which are characterized by an optimum growth temperature greater than 50 °C [19]. Few studies have examined the potential of thermotolerant microorganisms during anaerobic digestion. Suksong et al. [25] studied the gas production potential of thermotolerant microorganisms from the anaerobic digestion of oil palm empty fruit bunches. The maximum CH_4_ yields identified for *Clostridiaceae* and *Lachnospiraceae* with prehydrolysis empty fruit bunches were 252 mL CH_4_·g^−1^ vs. and 349 mL CH_4_·g^−1^ VS, respectively. Su et al. [20] discovered the potential of a thermotolerant methanotrophic consortium for producing methanol from biogas. To date, there have been no reports on the potential of thermotolerant microorganisms to produce biogas from the anaerobic digestion of anaerobic sludge.

Notably, investigation of biogas production and genomic analysis from the anaerobic digestion process under shifted-up temperatures has not been performed to date. Therefore, there is an urgent need to expand our understanding of this field. Owing to the heat-resistant characteristics and possible benefits of AD in WWTP-scale applications, further research regarding thermotolerant microorganisms needs to be performed, especially to investigate their potential for biogas production. Therefore, the objectives of this study were to investigate the potential of biogas production (especially CH_4_) and to identify the most biogas-producing microorganisms among the cultures after being shifted to several temperature levels.

## 2. Materials and Methods

### 2.1. Inoculum and Substrates

The inoculum (I) originated from anaerobically digested sludge obtained from a municipal sewage treatment plant located in Ube City, Yamaguchi Prefecture, Japan. The inoculum characteristics are listed in Table 1. The inoculum samples were then mixed with the substrate (S) solution before being placed in an incubator and exposed to gradually elevated temperature conditions. The substrate used for biogas production in this research was a glucose-based synthetic wastewater consisting of 1.5 g·L^−1^ glucose, 2 mg·L^−1^ NaHCO_3_, 2 mg·L^−1^ K_2_HPO_4_, 1 g·L^−1^ yeast extract, 0.7 g·L^−1^ (NH_4_)_2_HPO_4_, 0.75 g·L^−1^ KCl, 0.85 g·L^−1^ NH_4_Cl, 0.42 g·L^−1^ FeCl_3_·6H_2_O, 0.82 g·L^−1^ MgCl_2_·6H_2_O, 0.25 g·L^−1^ MgSO_4_·7H_2_O, 0.018 g·L^−1^ CoCl_2_·6H_2_O, and 0.15 g·L^−1^ CaCl_2_·2H_2_O. Glucose was chosen as the ideal carbon source for microbial metabolic transformations in the fermentation process, and is also a readily biodegradable substance abundantly found in municipal wastewater [26,27]. To obtain the maximum CH_4_ potential, the appropriate ratio between the microorganisms and substrate must be determined [28]. The CH_4_ yield, in theory, is independent of the inoculum to substrate ratio (I·S^−1^), and the I·S^−1^ ratio should influence only the kinetics of CH_4_ production [29].

In contrast, previous studies have demonstrated that the I·S^−1^ ratio can impact both the CH_4_ yield and production rate, as significant evidence suggested that the ratio directly influences the microorganism growth pattern [30,31,32]. Referring to the German Standard VDI4630, the I·S^−1^ ratio should be adjusted to more than two [33]. Other studies have discovered that a higher I·S^−1^ ratio generates more biogas on a consistent basis during the AD process, while a lower I·S^−1^ ratio produces less biogas due to the lower pH and accumulation of volatile fatty acids (VFAs) [34,35]. In this study, the I·S^−1^ ratio maintained at approximately 3.0, which indicates that 1 mL of substrate was added for every 3 mL of inoculum.

### 2.2. Experimental Procedures

Laboratory-scale anaerobic digester containers were later defined as vials. A total volume of 160 mL was prepared. To ensure obligate anaerobic conditions, the vial was filled with pure nitrogen gas to flush the remaining oxygen. Subsequently, the vials were capped with butyl rubber stoppers and aluminum caps. Since the laboratory-scale vials were used as the reactor in this research, there were some potential risks that may emerge, such as lower capacity to contain total biogas production, low sample provision to perform several monitoring parameters, and possibility that the sensitivities and instabilities in the laboratory scale reactors do not represent that in the full-scale digesters [36]. This research was divided into two categories: temperature shifted-up (shift-up) and controlled temperature condition. For the shift-up research, in phase 1, batch experiments were performed by adding 110 mL of sludge as inoculum to a vial mixed with 40 mL of substrate. The mixture was then incubated at 42 °C with shaking at 50 rpm for the first two weeks. This action was intended to enhance the growth of microorganisms and ensure the availability of nutrients during acclimatization.

After the first two weeks, phase 2 was initiated. Every time gas production declined sharply, up to 2 mL of substrate with a chemical oxygen demand (COD) of 2000 mg·L^−1^ was injected into the vial. This treatment altered the reactor system from a batch reactor to a fed-batch reactor system. The incubation was continued for 70 days of incubation period in the fed-batch reactor system. Every 70 days, the temperature was increased by 3 °C for the shift-up research until it reached 45 °C and 48 °C. Meanwhile, the controlled temperature research was carried out at 45 °C and 48 °C from the beginning of incubation without any temperature shifts. This study was intended to compare the biogas production and microbial communities among the shifted condition and stabilized condition. During the fermentation period, the total gas volume and composition were measured daily using gas chromatography.

A glass syringe was used to measure the volume of the biogas produced. The gas composition of the samples, such as H_2_, N_2_, CH_4_, and CO_2_, was determined using gas chromatography (GC-8APT/TCD; Shimadzu Co., Kyoto, Japan) with 60/80 activated charcoal mesh column (1.5 m × 3.0 mm internal diameter) and argon gas as the carrier gas. During operation, the temperatures of the injector, column, and detector were adjusted to 50 °C, 60 °C, and 50 °C, respectively.

### 2.3. Deoxyribonucleic Acid (DNA) Extraction and Sequencing

DNA was extracted according to the NucleoSpin^®^ soil manual. Sludge samples were prepared using an MN Bead Tube Type A (MACHEREY-NAGEL GmbH & Co., Düren, Germany). KG was mixed with lysis buffer SL1 and lysed using Enhancer SX. Contaminants were precipitated using lysis buffer SL3, and the lysate was filtered using a NucleoSpin^®^ Inhibitor Removal Column. Subsequently, the binding conditions were adjusted using Binding Buffer SB. DNA was bound by loading 550 µL sample on the NucleoSpin^®^ Soil Column. After the binding phase, the silica membrane was washed with binding buffer SB, wash buffer SW1, and SW2. Finally, the DNA was eluted using SE elution buffer. DNA samples were then delivered to the Faculty of Medicine, Yamaguchi University, Japan, for next-generation sequencing.

Next generation sequencing (NGS) was performed to acquire a broad range of genes or gene regions from phylum to genus using the 16 s ribosomal ribonucleic acid (RNA) gene amplicons for the Illumina MiSeq System, wherein DNA or RNA are sequenced using hybrid capture or amplicon-based approaches (previously transcribed into complementary DNA). Using these approaches, the genome (all 3 billion base pairs), all coding genes (exome; 1% of the genome or 30 million base pairs—that is 20,000 genes made of 180,000 exons), all RNA produced from genes (transcriptome), and any subset of these can be sequenced [37].

### 2.4. Microbial Diversity Analysis

Diversity index analysis was conducted to determine possible changes in the microbial communities during the anaerobic digestion process. A diversity index is a numerical measure of how many distinct types (such as species) are present in a dataset (a community), as well as the evolutionary relationships among individuals dispersed throughout those types, such as richness, divergence, and evenness [38]. In this study, Simpson’s diversity index, Shannon’s diversity index, and Shannon’s equitability index were utilized.

## 3. Results

### 3.1. Biogas Production under Shifted-Up Temperature

The fluctuation in daily CH_4_ production and cumulative CH_4_ production during the first incubation at 42 °C is shown in Figure 1. During the batch anaerobic digestion period, biogas production increased gradually with the cumulative volume of biogas generated being up to 316.5 mL on day 14, with cumulative CH_4_ production being up to 120.76 mL CH_4_. The CH_4_ content in the biogas increased rapidly in the first 8 days, reaching 69% on the 8th day, and 64.7% on average until day 13. Cumulative CH_4_ production increased gradually in the first 3 days, and then showed a substantial increase on day 7 (66.59 mL CH_4_). On the final day of the batch period, the cumulative methane production reached 120.76 mL CH_4_ which was equal to 1.43 L·g^−^^1^ COD feed. Theoretically, the energy recovery (expressed as CH_4_ production) from digested wastewater sludge through anaerobic process was 0.38 L·g^−1^ COD [39]. From this study, it took approximately four days of anaerobic digestion to exceed the theoretical CH_4_ production with the aforementioned inoculum and substrates, with an I·S^−^^1^ ratio of 2.75.

However, on the 14th day, methane production sharply decreased to 0 mL. Declining CH_4_ production indicates a dead phase of methanogenic activity [40]. This drawdown in CH_4_ production may also be linked to a decrease in pH caused by the interaction of VFAs with other fragmented precursors during oxidative processes [41]. To maintain CH_4_ production, 2 mL of the substrate was injected into the vial when CH_4_ production started diminishing due to the scarcity of nutrients. As illustrated in Figure 1a, substrate injection led to a spike in CH_4_ production as the activity of methanogenic microorganisms increased due to the availability of glucose as a carbon source and other nutrients that expedited microbial growth. Consequently, as shown in Figure 1a*, the cumulative volume of methane produced increased significantly after substrate addition. At the end of the incubation period at 42 °C, the cumulative CH_4_ production was observed to increase to 454.5 mL CH_4_, with a yield of 4.55 L·g^−1^ COD feed.

After incubation for 70 days at 42 °C, the incubation temperature was shifted to 45 °C with the same treatment conditions and hydraulic retention times. Even though the same sample was utilized, the cumulative CH_4_ calculation was restarted from 0 mL CH_4_. As presented in Figure 1b, the daily CH_4_ production peaked at 17.2 mL CH_4_ which was obtained on day 60 after receiving the 4th feed. From the Figure 1b*, the cumulative CH_4_ produced and yield after 70 days of incubation was 152.2 mL CH_4_ and 1.52 L·g^−1^ COD feed, respectively. This was less than the volume of CH_4_ produced during incubation at 42 °C by a factor of three. There was also a significant difference in CH_4_ production behavior after the temperature was increased to 45 °C, e.g., a shorter period required to shift from peak days to trough days, which denotes faster methanogenesis and death phase for methanogenic bacteria. This shorter methanogenesis phase can be attributed to the higher concentration of CO_2_ produced during the incubation period.

As shown in Figure 1c, CH_4_ production decreased further when the temperature was increased to 48 °C. After 70 days of incubation, only 86.57 mL of CH_4_ was produced, with a CH_4_ yield of 0.94 L·g^−1^ COD feed. The CH_4_ production trend after the 3rd feed at shifted-up 45 °C is a good illustration of the potential inhibition of methanogenic bacterial activity by the presence of high CO_2_ levels (Figure 2). The high CO_2_ levels indicated the occurrence of acetoclastic methanogenesis in the AD process, which later led to the abundant presence of VFAs, particularly acetic acid [42]. The decrease in CH_4_ was also parallel to the decrease in overall biogas production consisting of H_2_, N_2_, CH_4_ and CO_2,_ which is illustrated in Figure 3a. The total biogas production decreased from 1161.93 mL (11.61 L·g^−1^ COD feed) at 42 °C to 672 mL (6.7 L·g^−1^ COD feed) and then to 505 mL (5.49 L·g^−1^ COD feed) after the temperature was shifted to 45 °C and 48 °C, respectively. The decreasing CH_4_ production volume after being shifted-up to the higher temperature at 45 °C and 48 °C was followed by the decline in CH_4_ content in biogas compositions as seen in Figure 3b–d.

At the end of incubation period, the concentration of COD, total suspended solids (TSS), volatile suspended solids (VSS), and pH were measured for each temperature condition. As seen in Table 2, the concentration of COD was increasing up to 3-fold, in contrast to TSS and VSS that significantly decreased every temperature shift. This finding indicates that the number of microbial communities (represented by VSS) declined every upshifted thermal condition and subsequently causing a depletion on microbial activity, which eventually resulted in the lower organic matter consumed by microorganisms in the reactor. The pH is another parameter that affects digestion process. The pH increased substantially from 7.64 at 42 °C to 8.20 and 8.33 when the temperature was shifted-up to 45 °C and 48 °C, respectively. Subsequently, the biogas production decreased along with the rising pH. This finding supports a research study from Kouzi et al. [43] who discovered that the optimum pH range for sewage sludge AD was 7.0, while the biogas production was considerably lower in the reactors with higher pH of 8.0, 9.0, and 10.0.

### 3.2. Alpha Diversity Analysis

To study the changes in microbial diversity at the upshift temperature, samples from the reactor were used for DNA isolation for bioinformatic analysis using NGS. For comparison, several samples from other fed-batch reactors with controlled temperatures of 42 °C, 45 °C, and 48 °C were examined for their microbial diversity. This action was intended to elucidate the differences among microbial communities and compositions that matured under controlled and shifted-up temperatures. The Shannon diversity index (SDI) and Simpson index were used to measure and compare the richness of the microbiota at a certain temperature, while the Shannon equitability index (SEI) was assigned to approximate the evenness of the microbiota diversity.

As illustrated by Figure 4a,b, the index value for both richness and evenness of the microbiota communities spread within the range 3.2–3.7 and 0.46–0.54, respectively. The Simpson index, as seen in Figure 4c, ranged from 0.89 to 0.96, indicating high diversity for all sample conditions. The deviation of the diversity index between the shifted-up temperature conditions and controlled temperature conditions was not significantly discerned, which signifies that each reactor has a close similarity of microbiota abundance and species to each other. However, compared to the controlled temperature conditions, the shifted-up temperature conditions showed a significant drop in microbiota diversity, with an increase in temperature. The diversity index value declined from 3.72 at 42 °C to 3.22 at 48 °C. This indicated that several bacteria communities were vanished during the temperature shift. This was also confirmed by the decrease in the equitability index value; however, since the equitability values were greater than 0.1, some microorganism colonies managed to acclimatize to this chaotic condition and experienced massive growth while the other colonies became extinct. The effects of temperature on diversity were confirmed using analysis of variance (ANOVA), with significance of *p* < 0.05. The probability value (*p*-value) for all diversity indices (SDI, SEI, and Simpson index) was *p* < 0.001, which signified that there were statistically significant differences in relative abundances and diversity indices between several temperature conditions.

### 3.3. Microbial Community Structure

Overall, the number of methanogens decreased sharply when the temperature was shifted from 42 °C to 45 °C, as shown in Figure 5. *Methanosaeta* was the most dominant methanogenic archaea that existed during incubation at 42 °C, shifted up to 45 °C, subsequently shifted up to 48 °C, and also abundant during incubation atcontrolled temperatures of 45 °C and 48 °C. This result implied that *Methanosaeta* is a thermotolerant methanogen. The relative abundances of methanogens at the order level at various temperatures are shown in Figure 6a. The composition of methanogens in both shifted-up temperature and controlled temperature conditions was dominated by the orders *Methanobacteriales*, *Methanomicrobiales*, and *Methanosarcinales*. Among the three methanogens, *Methanosarcinales* was the most abundant (86.89% at 42 °C, 88.84% at shifted-up 45 °C, 59.14% at shifted-up 48 °C, 85.56% at controlled 45 °C, and 78.54% at controlled 48 °C).

At the family level, as illustrated in Figure 6b, the methanogen communities were composed of *Methanosaetaceae*, *Methanomicrobiaceae*, *Methanoregulaceae*, *Methanobacteriaceae*, *Methanosarcinaceae*, and *Methanospirillaceae*. The *Methanosaetaceae* family was abundant, with relative abundances of 85.01% at 42 °C, 88.05% at shifted-up 45 °C, 56.52% at shifted-up 48 °C, 84.76% at controlled 45 °C, and 51.14% at controlled 48 °C. In this study, *Methanosaeta* was the only descendant of the *Methanosaetaceae* family. At the genus level, *Methanoculleus*, *Methanolinea*, *Methanobacterium*, *Methanobrevibacter*, *Methanosarcina*, *Methanothermobacter*, *Methanofollis*, *Methanosalsum*, *Methanogenium*, and *Methanolobus* were detected at all temperatures (Figure 6c). However, during the incubation at a controlled temperature of 48 °C, the dominance of *Methanosaeta* was lower than under shifted-up temperature (84% of the total methanogens at shifted-up 48 °C and 51% at controlled 48 °C) while *Methanosarcina* genes were detected up to 27% of the total methanogens. These findings confirmed the results of Figeac et al. [44] who discovered that the family *Methanosarcinaceae* was the most abundant acetotrophic archaea in the initial thermophilic inoculum, whereas the *Methanosaetaceae* family was mostly found in the initial mesophilic inoculum. Therefore, the population of *Methanosaeta* with an initial temperature of 48 °C was considerably lower than that in the upshift condition that was initially acclimatized under mesophilic conditions (at 42 °C). Apart from acetotrophic methanogens, hydrogenotrophic methanogens, such as members of genera *Methanobacterium*, *Methanobrevibacter,* and *Methanothermobacter*, started to grow significantly after the reactor temperature was shifted up to 48 °C. Hydrogenotrophic methanogens favor thermophilic conditions, whereas acetotrophic methanogens cannot resist high temperatures [45].

*Methanobacterium* communities grew from 2.5% to 7.4% of the total methanogen population, and *Methanobrevibacter* population ranged from 0.76% to 1.64% at a shifted-up temperature of 42–48 °C. This finding contradicts previous research reporting that *Methanobacterium* genera were mostly found at lower mesophilic temperature (24–35% at 35 °C and 32–45% at 37 °C) and eradicated with increasing the temperature to 55 °C [44,46]. *Methanobrevibacter* genera were also found to be the dominant methanogens at 24 °C and 35 °C and vanished at 55 °C [44]. However, the researchers did not examine the existence of *Methanobrevibacter* genera at higher mesophilic temperatures (42–48 °C). Judging from the results of previous studies that showed *Methanobacterium* and *Methanobrevibacter* were abundant in the lower mesophilic conditions, our research reported a significant spike in the population of those methanogens after shifting up temperatures to higher mesophilic conditions, signifying that *Methanobacterium* and *Methanobrevibacter* genera potentially have heat-resistant characteristics that allow them to compromise the staggering increase in temperature conditions. Lastly, *Methanothermobacter* population increased from 0.17% to 0.24% relative abundance at 42–48 °C. This is not surprising as *Methanothermobacter* genera is a thermophilic methanogen that dominated methanogenesis at temperatures of 50 °C and higher [47,48,49,50].

The distribution of non-methanogenic bacteria is also an important factor for determining the influence of several bacteria on CH_4_ production. As seen in Figure 7a, *Clostridia* and *Synergistia* were the most abundant bacteria at the order level, with respective relative abundances of 31.11% and 18.50% at 42 °C, 42.98% and 20.88% at shifted-up 45 °C, 24.38% and 34.83% at shifted-up 48 °C, 23.43% and 30.49% at 45 °C, and 27.88% and 28.17% at 48 °C. *Clostridia* was found to be dominant at both mesophilic (this research) and thermophilic temperatures (at 52 °C) [51], indicating that the *Clostridia* order belongs to thermotolerant bacteria. According to the experimental results, *Synergistia* were found at higher mesophilic temperatures, but in some cases, were also found abundantly at low temperatures of 20 °C [52]. This suggested that *Synergistia* was resistant to both low-and high-temperature environments, leading to the conclusion that temperature had a responsive connection with the microbial community structure. At the family level, *Anaerobaculaceae, Clostridiaceae,* and *Thermoanaerobacterceae* dominated the microbial communities, as shown in Figure 7b. At the genus level (see Figure 7c), *Anaerobaculum, Fervidobacterium, Tepidanaerobacter, Clostridium, Moorella, Aminiphilus, Carboxydocella*, and *Methanosaeta* are some microorganism genera that exhibited noteworthy growth during the temperature shift. *Anaerobaculum* and *Tepidanaerobacter* from the *Thermoanaerobacterales* family are syntrophic bacteria that have an essential role in converting short-chain fatty acids to methanogenic components such as acetate, H_2_, and formate [53]. *Moorella* has been identified as an acidogenic bacterium [54], whereas *Clostridium* is a hydrogen-producing bacterium that plays a key role in the hydrolysis process [46].

Compared to the other types of microorganisms, methanogenic archaea were shown to have the least portion of the population among the microbial communities. The decrease in the methanogen population accelerated faster in shifted-up temperature conditions than in controlled temperature conditions. However, the number of methanogen populations is unlikely to affect CH_4_ production. The cumulative CH_4_ production decreased along with the decline in the methanogen:total population (M·TP^−1^) ratio during the shifted-up temperature period. This result contradicts the outcome of the controlled temperature, in which the cumulative CH_4_ production increased conspicuously despite the fluctuation in the M·TP^−1^ ratio (Figure 8a). To date, no particular ratio has been found to be effective in understanding the influence of the microbial ratio (the existence of a particular microorganism) on biogas production. The closest ratio was that of sulphate-reducing bacteria (SRB) to methanogens (SRB·M^−1^). Previous research has stated that the existence of SRBs in the AD process may inhibit CH_4_ production as it would compete with methanogens for convenient H_2_, acetate, propionate, and butyrate [55]. From the shifted-up temperature experiment, the SRB·M^−1^ ratio showed harmony with the statement of previous research. The higher the SRB·M^−1^ ratio, the lower the CH_4_ production as the SRB emulated the methanogens in consuming available H_2_ (Figure 8b).

Nevertheless, the results from the controlled temperature experiments showed that the SRB·M^−1^ ratio was also ineffective in determining the relationship between the ratio and CH_4_ production. At controlled 48 °C, the maximum CH_4_ volume production was observed despite the low M·TP^−1^ ratio and high SRB·M^−1^ ratio. Under these conditions, the populations of *Methanosarcina* and *Methanoculleus* genera were the most abundant. *Methanosarcina* genera are known to be the major contributors to CH_4_ production [56] and manage to perform all methanogenesis pathways (hydrogenotrophic, acetoclastic, and methylotrophic) that help them survive in food competition [57]. In contrast to other methanogens, *Methanosarcina* was capable of growing significantly under high concentrations of VFAs and ammonia, which are the foremost inhibitors in biogas production [58]. *Methanoculleus*, in contrast, has been reported to increase in abundance along with elevated sulfate concentration [59]. Thus, both *Methanosarcina* and *Methanoculleus* acclimatized well and were attributed to high CH_4_ production under high inhibitor concentrations.

These findings may explain the phenomenon of increasing CH_4_ production at a controlled temperature of 48 °C. The influence of *Methanosarcina* and *Methanoculleus* on CH_4_ production was demonstrated by considering the decreasing volume of CH_4_ production parallel to the decrease in *Methanosarcina* and *Methanoculleus* abundance during shifts in temperature. *Methanosarcina* abundances decreased from 2% to 0.75% and 0.59%, parallel to the *Methanoculleus* population that declined from 6.52% to 4.96% and 3.81% during the incubation at temperature of 42 °C, shifted up to 45 °C, and shifted up to 48 °C, respectively.

## 4. Discussion

In AD processes, temperature has a significant influence on biogas production and microbial ecology [60,61]. There has been a number of research that examined the effects of temperature in mesophilic and thermophilic conditions [18,62,63]. However, to the best of our knowledge, the assessment of biogas production (especially CH_4_) and microbial community adaptation under multiple rising temperature conditions in a fed-batch reactor has not been widely studied. We expected instability in microbial communities (especially methanogens) and decreased CH_4_ production, along with the temperature shift process owing to perturbations caused by sudden temperature changes. Among the three temperature conditions, there was a noticeable decrease in CH_4_ production when the temperature was increased. The cumulative CH_4_ production decreased from 454 mL at 42 °C to 152 mL after increasing the temperature to 45 °C and to 86.57 mL after increasing the temperature to 48 °C. This result is consistent with previous findings for the shifted-up temperature [51,64]. Each temperature shift was conducted after 70-day incubation periods in order to allow a period of acclimatization for methanogens and other bacteria. The total methanogen abundance in shifted-up 45 °C and 48 °C were close to that in controlled 45 °C and 48 °C after 70 days operation. Previous studies showed potential steadier operation condition in term of CH_4_ production after acclimatization for 100–140 days, yet the risks of instability still exists [51,65].

Beale et al. [64] investigated the effect of upshift temperature shock from 37 °C to 42 °C on the biogas production volume of anaerobically digested sludge. Similarly, the biogas generated after the temperature was increased to 42 °C was persistently lower than that from the controlled digester at 37 °C during the first 32 days of operation. Identical results were also reported by Ziembinska-Buczynska et al. [66] who found a significant decrease in the biogas production rate from 70.5 L·day^−1^ to 28.6 L·day^−1^ along with an increase in temperature from 38 °C to 55 °C. Researchers also discovered that there was a decrease in microbiota diversity as the temperature of the digester influenced the evolution from mesophilic to thermophilic conditions. Some methanogens cannot survive at higher temperatures (heat unresistant), e.g., *Methanobrevibacter* (37–39 °C), *Methanogenium* (20–25 °C), or *Methanobacterium* (37–45 °C) [67]. However, our research contradicts the findings of Bouskova et al. [68] who discovered higher CH_4_ production after the temperature was shifted from 42 °C to 47 °C, 51 °C, and 55 °C. A possible reason for the observed discrepancies was the characteristics and ratio of the inoculum and substrates. The researchers used an inoculum with TS of 31.24 g·L^−1^ and vs. of 14.48 g·L^−1^. Meanwhile in our study, the inoculum consisted of 8 g·L^−1^ TS and 3 g·L^−1^ VS. Previous researchers also utilized a mixture of primary sludge and waste activated sludge as substrates which also contained seeds of microorganisms that maintained the longevity of biogas production.

We also found that the presence of excessive CO_2_ in the reactor may have led to lower methane production. Methanogenic bacteria require CO_2_ and H_2_ to produce CH_4_, which indicates that if the CO_2_ volume was greater than CH_4_ after substrate feeding, this signifies the failure of these bacteria to consume sufficient quantities of CO_2_ and H_2,_ which consequently would lead to the accumulation of VFAs, lower CH_4_ yield, and low pH [69,70]. Low pH is a serious concern as it inhibits methanogenic bacteria due to the increase in the concentration of free acid molecules, which is harmful for microorganisms and impacts enzymatic activity [71]. It has been suggested that microbiota composition and methanogenic pathways are altered when encountering an immediate low pH and high acetate crisis (pH 5.5–6.5, completely hindered at pH 5.0) [72].

Temperature shifts also affected the microbial communities in the reactor, especially the methanogens. The number of methanogens decreased significantly after the temperature was shifted from 42 °C to 45 °C and then stabilized at 48 °C, as shown in Figure 4. This instability supported the study by Westerholm et al. [51] who found that immense perturbation occurred in the interval of 40–44 °C, signifying that the 40–44 °C temperature range had a significant impact on both mesophilic and thermophilic microbial populations. The only conceivable interpretation for this phenomenon is that the temperature range may be greater than the upper threshold for the growth of mesophiles but not sufficiently high for the growth of thermophiles [73]. Because our reactor was initially developed using mesophilic anaerobic sludge, this potentially limits the abundance of thermophilic microorganisms. Tian et al. [74] omitted the 40–44 °C area and still found a transitory decrease in total methanogen concentration after the temperature was shifted (37–55 °C), but then recovered quickly on day 11.

Among all the methanogens, *Methanosaeta* (*Methanosaetaceae* family) was the most abundant under all temperature conditions in this study. This finding contradicts the findings of Kim et al. [75] who reported that the *Methanosaetaceae* family started to dominate the microbial structure only at temperatures above 45 °C. A plausible explanation for this difference is that the researchers started cultivation at 35 °C, which was more favorable to the growth of *Methanomicrobiales* order than *Methanosarcinales* (the ancestor of *Methanosaetaceae*). The instability of methanogen populations in anaerobic digestion also led to an increase in some types of bacteria that contributed to H_2_ and VFAs consumption, such as SRBs. As illustrated in Figure 7b, our research shows that the increasing SRB/methanogen ratio has a considerable influence on the decrease in CH_4_ production under the shifted-up temperature conditions. This result supports the idea from previous studies that reported that sulfide generation by SRBs inhibits methanogenesis, with the latter being the leading rival of methanogens for electron donors and substrates [65,76]. In addition, Beale et al. [64] emphasized that even a small amount of SRBs was enough to inhibit biogas production, as methanogens were vulnerable to the toxicity caused by metabolic products of SRBs.

Although this study characterized the behavior of some microorganisms and their influence on CH_4_ production during the shifted-up temperature, there are still several unidentified factors that can potentially affect the outcome of the AD process. Further studies are needed to determine and characterize the mechanism by which shifted temperature may affect the abundance of microorganisms, especially methanogenesis-related microbiota (e.g., methanogens, SRB, methanotrophs, hydrogenotrophs, acetotrophs, nitrogen-fixing bacteria, and sulfate-oxidizing bacteria) and the influence on CH_4_ production. Other microbial communities with syntrophic and fermentative behaviors, as well as their metabolic networks, merit study. Minimizing the number of unknown microorganisms may also provide a clearer insight into the relationship between the abundance of microorganisms and CH_4_ production, as in this study, we detected a large number of unknown bacteria.

## 5. Conclusions

Treating wastewater sludge using anaerobic digestion does not eliminate the risk of temperature instability. Consequently, the effects of shifting the temperature during anaerobic digestion of anaerobic sludge were investigated in this study. The results showed a considerable reduction in the CH_4_ cumulative gas production, from 454 mL (4.55 L·g^−1^ COD) to 152 mL (1.52 L·g^−1^ COD) then to 86.57 mL (0.94 L·g^−1^ COD) when the temperature of the reactor was increased from 42 °C to 45 °C and subsequently to 48 °C, respectively. Several factors have been attributed to the decrease in CH_4_ production under the shifted-up temperature, such as the decreasing methanogen population (expressed as the M·TP^−1^ ratio) due to intense food competition, increasing SRB populations over methanogens, and low abundance of major CH_4_ producers (e.g., *Methanosarcina* and *Methanoculleus*). *Methanosaeta* was the most dominant methanogen in this study, while *Anaerobaculum* and *Tepidanaerobacter* were the most abundant syntrophic bacteria, and *Clostridium* which are known as hydrogen-producing bacteria. Overall, the diversity of the anaerobic microbial consortium observed in this study altered slightly during the shift in the thermal conditions. This indicated that the majority of the communities belonged to thermotolerant microorganisms.

## Figures and Tables

**Figure 1 ijerph-19-05728-f001:**
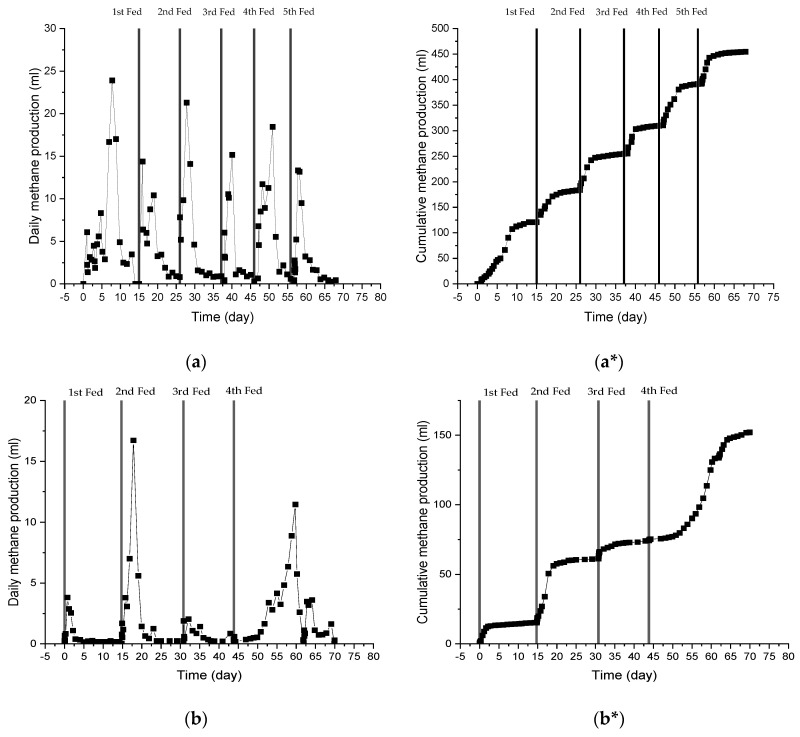
Daily (without *) and cumulative (with *) methane production of anaerobic sludge at (**a**,**a***) 42 °C, (**b**,**b***) shifted-up to 45 °C, and (**c**,**c***) shifted-up to 48 °C.

**Figure 2 ijerph-19-05728-f002:**
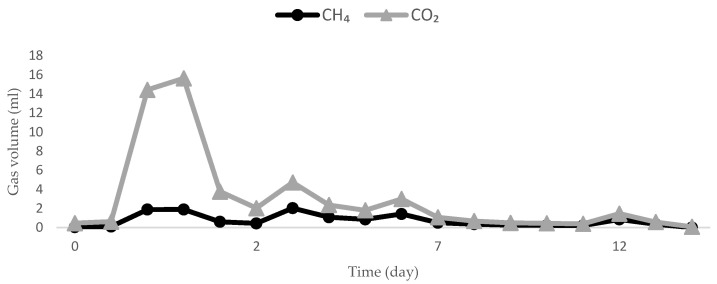
The production of CO_2_ after the 3rd feed compared to CH_4_ production in 12 consecutive days.

**Figure 3 ijerph-19-05728-f003:**
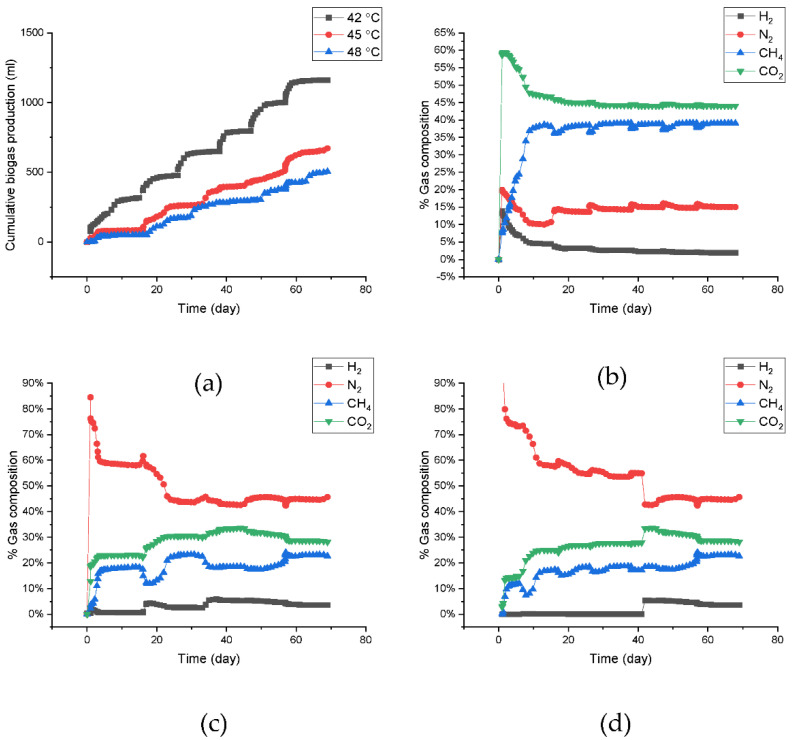
Total biogas production (H_2_, N_2_, CH_4_, and CO_2_) of anaerobic sludge during the shifted-up temperature conditions (**a**) and biogas composition at 42 °C (**b**), shifted-up to 45 °C (**c**), and shifted-up to 48 °C (**d**).

**Figure 4 ijerph-19-05728-f004:**
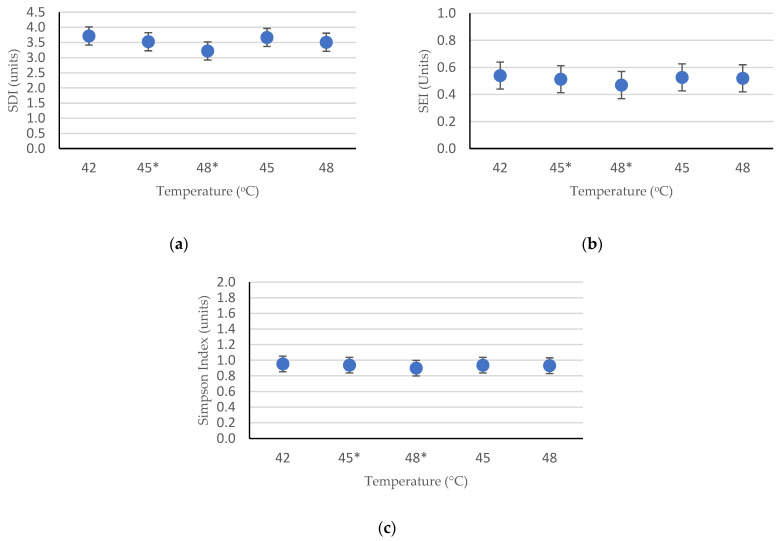
The richness and evenness index of the total microorganisms’ communities on each reactor: (**a**) Shannon-Wiener Diversity Index; (**b**) Shannon Equitability Index; and (**c**) Simpson Diversity Index. **Note:** (*) signs the shifted-up temperature.

**Figure 5 ijerph-19-05728-f005:**
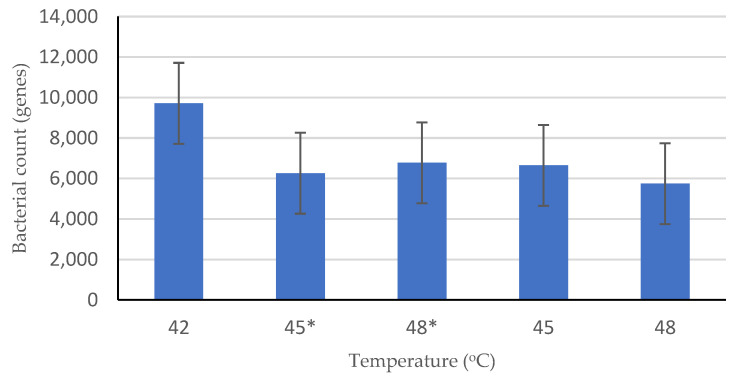
Methanogens total number of genes hit by NGS during anaerobic digestion process in several temperature conditions. **Note:** (*) signs the shifted-up temperature.

**Figure 6 ijerph-19-05728-f006:**
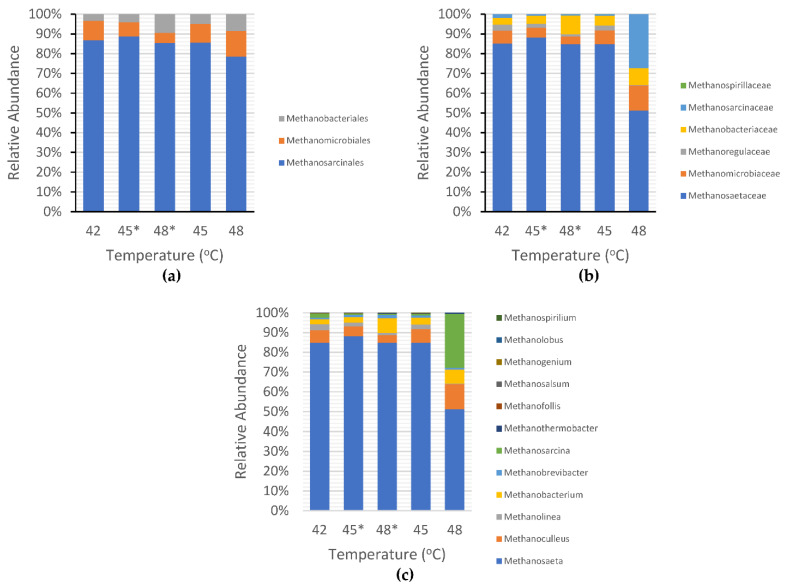
Methanogens distribution in shifted-up temperature (*) and controlled temperature: (**a**) order level; (**b**) family level; and (**c**) genus level.

**Figure 7 ijerph-19-05728-f007:**
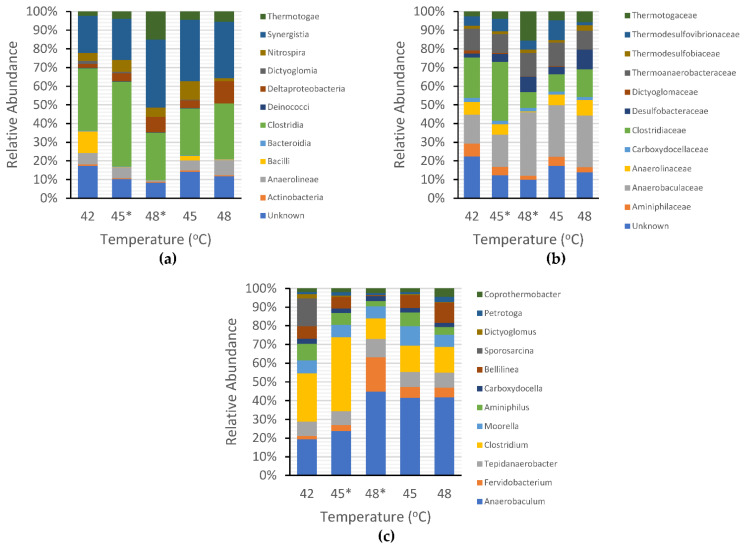
Bacteria (non-methanogens) community distribution in shifted-up temperature (*) and controlled temperature: (**a**) order level; (**b**) family level; and (**c**) genus level.

**Figure 8 ijerph-19-05728-f008:**
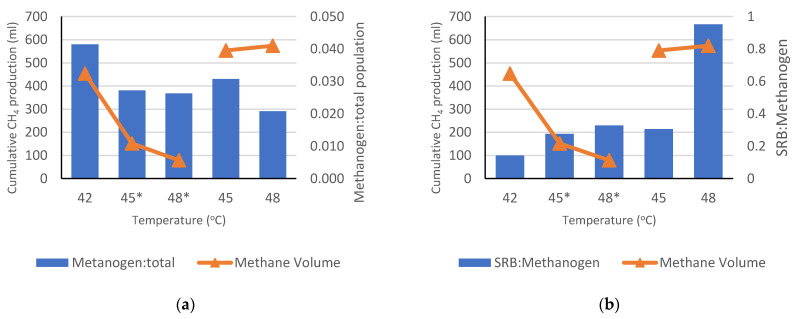
Comparison between cumulative CH_4_ production and M·TP^−1^ ratio (**a**) and sulphate-reducing bacteria (SRB) to methanogen ratio (**b**). **Note:** (*) signs the shifted-up temperature.

**Table 1 ijerph-19-05728-t001:** Characteristics of anaerobic sludge as inoculum.

Parameters	Anaerobic Sludge	Units
pH	7.09	pH = −log_10_[*a*(H^+^)]
Total Solid (TS)	8000	mg/L
Volatile Solid (VS)	3000	mg/L
Fixed Solid (FS)	5000	mg/L
VS/TS ratio	0.37	-

**Table 2 ijerph-19-05728-t002:** Effluent quality in each temperature condition after incubation period.

Temperature (°C)	pH	Total Suspended Solids (TSS) in mg·L^−1^	Volatile Suspended Solids (VSS) in mg·L^−1^	VSS.TSS^−1^	Chemical Oxygen Demand (COD) in mg·L^−1^
42 °C	7.64	7300	4675	0.64	498.72
45 °C	8.20	6860	4105	0.60	1911.76
48 °C	8.33	6880	3465	0.50	3690.52

## Data Availability

Not applicable.

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
