# Peer review of "Effects of Stepwise Temperature Shifts in Anaerobic Digestion for Treating Municipal Wastewater Sludge: A Genomic Study"

_ijerph, 2022, doi:10.3390/ijerph19095728_

Round 1

Reviewer 1 Report

Manuscript

Title: „Effects of Stepwise Temperature Shifts in Anaerobic Digestion for Treating Municipal Wastewater Sludge: A Genomic Study”

Authors: Gede Adi Wiguna Sudiartha, Tsuyoshi Imai, and Yung-Tse Hung

Dear Authors

I revised the manuscript: " Effects of Stepwise Temperature Shifts in Anaerobic Digestion for Treating Municipal Wastewater Sludge: A Genomic Study " submitted to the “International Journal of Environmental Research and Public Health” Journal. The paper is very interesting. However, I have some concerns, which need to be addressed.

Line 1-2. The theme of the article is logical and reflects well the issues addressed.  

Abstract:

Line 10-24

Line 19, 20. „….from 4.55 L/g COD…..” Indication of the unit of measure "L/g” using a fractional dash is acceptable but is colloquial in meaning. However, exponential notation should be used.

Please use the notation of the quotient in units of measure using mathematical notation with a power exponent for example: L·g-1

Line 20 „….(M/TP) ratio of 0.041…..” Indication of the unit of measure "Methanogen/Total population” using a fractional dash is acceptable but is colloquial in meaning. However, exponential notation should be used.

Please use the notation of the quotient in units of measure using mathematical notation with a power exponent for example: M·TP-1.

The introduction of the genesis of the topic and the rationale is correct and does not dominate the content of the abstract. The goal and scope of the research are presented in a very laconic way. The research methodology is introduced through information about the realization of the main theses of the work. In general, the abstract is understandable for the reader. However, the abstract does not reflect the multi-threaded structure of the scientific work. Lack of direct reference to sewage sludge as a test substrate. Please take this into account.

„COD” – Please explain the introduced abbreviations of terms, possibly as soon as they are introduced in the content of the article. A single explanation is sufficient in the execution of the explanation.

For example:

COD – chemica oxygen demand

Line 150. „chemical oxygen demand (COD) of 2000 mg/l was…” The explanation of the abbreviated name appears first on line 150. Authors may leave the explanation on line 150 or accelerate the explanation on line 19. A single explanation of the abbreviated term in the text of the article is sufficient.

Line 19, 21. „….at 42°C…..” The unit of measure and value should be separated by a space.

Keywords are chosen correctly. Multiple keywords is not a mistake but requires an editor's opinion. Please take this into consideration.

  1. Introduction

The rationale for undertaking the research topic is logical and comprehensively documented. The order of citing literature sources in the chapter is correct. No critical comments on the scientific level of the chapter.

Line 61. „….342.0 mL CH4/g VS…..” Indication of the unit of measure "mL CH4/g VS” using a fractional dash is acceptable but is colloquial in meaning. However, exponential notation should be used.

Please use the notation of the quotient in units of measure using mathematical notation with a power exponent for example: mL CH4·g-1 VS.

We should mark the „litre” with a small „l”.

Line 62. „……and 179.3 mL H2/g VS……” Indication of the unit of measure "mL H2/g VS” using a fractional dash is acceptable but is colloquial in meaning. However, exponential notation should be used.

Please use the notation of the quotient in units of measure using mathematical notation with a power exponent for example: mL H2·g-1 VS.

We should mark the „litre” with a small „l”.

Line 63. „….biogas at 230 ± 29 ml/l/d….” Indication of the unit of measure "ml/l/g” using a fractional dash is acceptable but is colloquial in meaning. However, exponential notation should be used.

Please use the notation of the quotient in units of measure using mathematical notation with a power exponent for example: ml·l-1·g-1

Line 66, 69, 70, 88. „…(<30°C)….” The sign of the mathematical operation and the numerical value are separated by a space.

The unit of measure and value should be separated by a space.

Line 66, 85. „….(30–40°C)…..” The unit of measure should be present with both range values.

Line 71, 72. „….was 0.246 m3 CH4/m3 per digester….” Indication of the unit of measure "m3 CH4/m3 ” using a fractional dash is acceptable but is colloquial in meaning. However, exponential notation should be used.

Please use the notation of the quotient in units of measure using mathematical notation with a power exponent for example: m3 CH4·m-3 .

Line 74. „…..(0.56–0.70 L CH4/g VS and 0.25–0.32 L CH4/g VS respectively……”

Indication of the unit of measure "L CH4/g VS” using a fractional dash is acceptable but is colloquial in meaning. However, exponential notation should be used.

Please use the notation of the quotient in units of measure using mathematical notation with a power exponent for example: L CH4·g-1 VS.

We should mark the „litre” with a small „l”.

The unit of measure should be present with both range values.

Line 92. „….were 252 and 349 ml CH4/g VS…..” Indication of the unit of measure "mL CH4/g VS” using a fractional dash is acceptable but is colloquial in meaning. However, exponential notation should be used.

Please use the notation of the quotient in units of measure using mathematical notation with a power exponent for example: ml CH4·g-1 VS.

The unit of measure should be present with both range values.

  1. Materials and Methods

The methodological assumptions and selection of research tools are correct. The methodological assumptions of the study are well considered and well documented.

There is a lack of information on the moment of application and the importance of dividing the research into the part: with shifted temperature and controlled temperature. Please clearly define both research situations already in chapter 2. 

Line 113, 114, 115, 116, 150. „….of 1.5 g/L glucose, 2 mg/L…..” Indication of the unit of measure „mg/L”,  "g/L” using a fractional dash is acceptable but is colloquial in meaning. However, exponential notation should be used.

Please use the notation of the quotient in units of measure using mathematical notation with a power exponent for example: mg·L-1, g·L-1 .

Line 115, 116. „….FeCl3.6H2O….” I propose a clear mathematical notation with a dot, for example FeCl3·6H2O

Line 120-121, 127, 128, 129, 130. „…..inoculum to substrate ratio (I/S), and the I/S ratio…..”

Inoculum (I), substrates (S), please complete the abbreviations (symbols) in the text.

Please use the notation of the quotient in units of measure using mathematical notation with a power exponent for example: I·S-1

Line 122. Table 1. The unit of measurement for pH is pH= –log10[a(H+)]

Line 130-131. „….approximately 3 ml/1 ml…..” The notation with a slash as a symbol for division (ratio symbol) is rather unsuggestive and colloquial.

Line 145, 148, 153, 154, 155, 162, 163. „….at 42°C with…” The unit of measure and value should be separated by a space.

Line 172. „…DNA….” Please explain the introduced abbreviations of terms, possibly as soon as they are introduced in the content of the article. A single explanation is sufficient in the execution of the explanation.

For example:

DNA: genetic code, Deoxyribonucleic acid?

Line 176. „….RNA…” Please explain the introduced abbreviations of terms, possibly as soon as they are introduced in the content of the article. A single explanation is sufficient in the execution of the explanation.

For example:

RNA: genetic code, ribonucleic acid (RNA)

Line 178. „….into cDNA….” Please insert an explanation in the text, for example "Complementary DNA"

  1. Results

The research findings are evidently multithematic in nature. There is a lack, in earlier chapters, of sufficient announcement of research themes. In its present form, the research findings are surprising due to the lack of preparation of an adequate research scope. Please take this into account.

Presentation of results logical and correct.

Line 202-203. „….inoculum and substrates, with an I/S ratio of 2.75…..”

Inoculum (I), substrates (S), please complete the abbreviations (symbols) in the text.

Please use the notation of the quotient in units of measure using mathematical notation with a power exponent for example: I·S-1

Line 215. „….4.55 L CH4/gCODadded. …….” Indication of the unit of measure "CH4/g” using a fractional dash is acceptable but is colloquial in meaning. However, exponential notation should be used.

Please use the notation of the quotient in units of measure using mathematical notation with a power exponent for example: CH4·g-1 .

Please note that a space is needed between the unit of measure "g" and the parameter description "COD". Please correct this.

CODadded. Please explain the abbreviation „added” in the bottom apostrophe.

Line 199, 201, 221, 230, 236, 237. „….5.49 L/gCOD…” Indication of the unit of measure "L/g” using a fractional dash is acceptable but is colloquial in meaning. However, exponential notation should be used.

Please use the notation of the quotient in units of measure using mathematical notation with a power exponent for example: L·g-1

Please note that a space is needed between the unit of measure "g" and the parameter description "COD". Please correct this.

Line 250-251. „…Figure 2. The production of CO2 was spiking after the 3rd fed while the CH4 production was below 4 mL CH4 in 12 consecutive days. ….”

The name of the figure is more a commentary than information about what characterises the figure. Please correct this.

Line 268. Please repeat the explanation of the abbreviated names in Table 2:

TSS – total suspended solids

VSS – volatile suspended solids

VSS/TSS ratio - Please use the notation of the quotient in units of measure using mathematical notation with a power exponent for example: VSS·TSS-1.

Line 272 „….DNA….”. Please ensure that the abbreviated name is explained beforehand.

Line 294. "ANOVA" please explain the abbreviation, for example (analysis of variance)

Line 294, 295. „P-value”, please explain the abbreviation, for example (probability value)

Line 338-340. Figure 5. A better designation of the ordinate axis is "Bacterial count" instead of "Methanogenic bacteria". The axis designation should be unambiguous.

Line 341 -344. Figure 6. a), b) c). Lack of unit of measure for the temperature parameter for the „abscissa” axis (OX axis).

Line 348, 349, 350, 353, 360. „….from 0.17 to 0.24% relative abundance at 42–48°C. ….”

The unit of measure and value should be separated by a space (0C).

The unit of measure should be present with both range values.

Line 416 Figure 7 a), b). Lack of unit of measure for the temperature parameter for the „abscissa” axis (OX axis).

Line 399, 402, 404, 424. Figure 8. „SRB/Methanogen”. Indication of the unit of measure "SRB/Methanogen” using a fractional dash is acceptable but is colloquial in meaning. However, exponential notation should be used.

Please use the notation of the quotient in units of measure using mathematical notation with a power exponent for example: SRB·Methanogen-1, SRB·M-1 .

Line 389, 392, 404, 424. „….M/TP ratio…..” Indication of the unit of measure "Methanogen/Total population” using a fractional dash is acceptable but is colloquial in meaning. However, exponential notation should be used.

Please use the notation of the quotient in units of measure using mathematical notation with a power exponent for example: M·TP-1.

Line 430-431. „…..from 2–0.75–430 0.59% and 6.52–4.96–3.81%.....”

The range notation is ambiguous. Please use a more intuitive form of presenting the results.

Line 192, 214, 216, 222, 224, 229, 236, 238, 247, 263, 264, 274, 288, 289, 299, 300, 301, 306, 307, 308, 312, 313, 318, 319, 324, 326, 329, 349, 350, 351, 352, 362, 366, 367, 371, 369, 427, 431, 432. „….42°C…..” The unit of measure and value should be separated by a space.

  1. Discussion

Discussion of the results conducted correctly based on the discussion of own results and the current state of knowledge. Multiple attempts to explain the findings indicate the proficiency of the authors in the research topic.

Line 457. „….70.5 L/day to 28.6 L/day…..” Indication of the unit of measure "L/day” using a fractional dash is acceptable but is colloquial in meaning. However, exponential notation should be used.

Please use the notation of the quotient in units of measure using mathematical notation with a power exponent for example: L·day-1.

We should mark the „litre” with a small „l”.

Line 466, 467. „….31.24 g/l…..” Indication of the unit of measure "g/l” using a fractional dash is acceptable but is colloquial in meaning. However, exponential notation should be used.

Please use the notation of the quotient in units of measure using mathematical notation with a power exponent for example: g·l-1.

Line 130, 206, 234, 410, 474. „….VFAs…..” The abbreviation was described on 130 lines, correctly.

Line 482. „….42 to 45°C….” The unit of measure should be present with both range values. 

Line 458, 461, 462, 484, 490, 491. „…37–55°C…” The unit of measure should be present with both range values. 

Line 444, 445, 448, 449, 452, 455, 464, 482, 496, 497. „….35°C…..” The unit of measure and value should be separated by a space.

Line 500. „….VFA consumption….” The abbreviation was explained as VFAs meaning plural. Please explain the abbreviation again for a correct understanding of the article content.

  1. Conclusions

Conclusions based on the research results are justified.

Line 525 – 526. „….(4.55 L/gCOD) to 152 ml (1.52 L/gCOD) then to 86.57 ml (0.94 L/gCOD)……” Indication of the unit of measure "L/gCOD” using a fractional dash is acceptable but is colloquial in meaning. However, exponential notation should be used.

Please use the notation of the quotient in units of measure using mathematical notation with a power exponent for example: L·g-1 COD.

The name of the parameter (COD) and its unit of measurement (g) should be separated by a space.

86 ml → 0.94 L please verify dissonance of values. 

Line 527. „….42°C…..” The unit of measure and value should be separated by a space.

Author Response

Response to Reviewer 1 Comments

Dear reviewer, we would like to thank you for the noteworthy feedbacks. We appreciate the time and effort that you have dedicated to providing valuable feedback on my manuscript. We have been able to incorporate changes to reflect most of the suggestions provided by the reviewer. We have highlighted the changes in the manuscript using Track Changes function according to Editor’s suggestion.

Here is a point-by-point response to the reviewer’s comments and concerns. The letters written on black ink are the concerns, comments, and questions. While the letters written on red ink are the responses from us as the authors.

  • Point 1: I revised the manuscript: " Effects of Stepwise Temperature Shifts in Anaerobic Digestion for Treating Municipal Wastewater Sludge: A Genomic Study " submitted to the “International Journal of Environmental Research and Public Health” Journal. The paper is very interesting. However, I have some concerns, which need to be addressed.Line 1-2. The theme of the article is logical and reflects well the issues addressed.  

Response 1: Thank you very much for the notable comments.

  • Point 2: Line 19, 20. „….from 4.55 L/g COD…..” Indication of the unit of measure "L/g” using a fractional dash is acceptable but is colloquial in meaning. However, exponential notation should be used. Please use the notation of the quotient in units of measure using mathematical notation with a power exponent for example: L·g-1

Line 20 „….(M/TP) ratio of 0.041…..” Indication of the unit of measure "Methanogen/Total population” using a fractional dash is acceptable but is colloquial in meaning. However, exponential notation should be used.

Please use the notation of the quotient in units of measure using mathematical notation with a power exponent for example: M·TP-1.

Line 61. „….342.0 mL CH4/g VS…..” Indication of the unit of measure "mL CH4/g VS” using a fractional dash is acceptable but is colloquial in meaning. However, exponential notation should be used.

Please use the notation of the quotient in units of measure using mathematical notation with a power exponent for example: mL CH4·g-1 VS.

We should mark the „litre” with a small „l”.

Line 62. „……and 179.3 mL H2/g VS……” Indication of the unit of measure "mL H2/g VS” using a fractional dash is acceptable but is colloquial in meaning. However, exponential notation should be used.

Please use the notation of the quotient in units of measure using mathematical notation with a power exponent for example: mL H2·g-1 VS.

We should mark the „litre” with a small „l”.

Line 63. „….biogas at 230 ± 29 ml/l/d….” Indication of the unit of measure "ml/l/g” using a fractional dash is acceptable but is colloquial in meaning. However, exponential notation should be used.

Please use the notation of the quotient in units of measure using mathematical notation with a power exponent for example: ml·l-1·g-1

Line 66, 69, 70, 88. „…(<30°C)….” The sign of the mathematical operation and the numerical value are separated by a space.

The unit of measure and value should be separated by a space.

Line 66, 85. „….(30–40°C)…..” The unit of measure should be present with both range values.

Line 71, 72. „….was 0.246 m3 CH4/m3 per digester….” Indication of the unit of measure "m3 CH4/m3 ” using a fractional dash is acceptable but is colloquial in meaning. However, exponential notation should be used.

Please use the notation of the quotient in units of measure using mathematical notation with a power exponent for example: m3 CH4·m-3 .

Line 74. „…..(0.56–0.70 L CH4/g VS and 0.25–0.32 L CH4/g VS respectively……”

Indication of the unit of measure "L CH4/g VS” using a fractional dash is acceptable but is colloquial in meaning. However, exponential notation should be used.

Please use the notation of the quotient in units of measure using mathematical notation with a power exponent for example: L CH4·g-1 VS.

We should mark the „litre” with a small „l”.

The unit of measure should be present with both range values.

Line 92. „….were 252 and 349 ml CH4/g VS…..” Indication of the unit of measure "mL CH4/g VS” using a fractional dash is acceptable but is colloquial in meaning. However, exponential notation should be used.

Please use the notation of the quotient in units of measure using mathematical notation with a power exponent for example: ml CH4·g-1 VS.

The unit of measure should be present with both range values.

Line 113, 114, 115, 116, 150. „….of 1.5 g/L glucose, 2 mg/L…..” Indication of the unit of measure „mg/L”,  "g/L” using a fractional dash is acceptable but is colloquial in meaning. However, exponential notation should be used.

Please use the notation of the quotient in units of measure using mathematical notation with a power exponent for example: mg·L-1, g·L-1 .

Line 120-121, 127, 128, 129, 130. „…..inoculum to substrate ratio (I/S), and the I/S ratio…..”

Please use the notation of the quotient in units of measure using mathematical notation with a power exponent for example: I·S-1

Line 145, 148, 153, 154, 155, 162, 163. „….at 42°C with…” The unit of measure and value should be separated by a space.

Line 202-203. „….inoculum and substrates, with an I/S ratio of 2.75…..”

Inoculum (I), substrates (S), please complete the abbreviations (symbols) in the text.

Please use the notation of the quotient in units of measure using mathematical notation with a power exponent for example: I·S-1

Line 215. „….4.55 L CH4/gCODadded. …….” Indication of the unit of measure "CH4/g” using a fractional dash is acceptable but is colloquial in meaning. However, exponential notation should be used.

Please use the notation of the quotient in units of measure using mathematical notation with a power exponent for example: CH4·g-1 .

Please note that a space is needed between the unit of measure "g" and the parameter description "COD". Please correct this.

Line 199, 201, 221, 230, 236, 237. „….5.49 L/gCOD…” Indication of the unit of measure "L/g” using a fractional dash is acceptable but is colloquial in meaning. However, exponential notation should be used.

Please use the notation of the quotient in units of measure using mathematical notation with a power exponent for example: L·g-1

Please note that a space is needed between the unit of measure "g" and the parameter description "COD". Please correct this.

VSS/TSS ratio - Please use the notation of the quotient in units of measure using mathematical notation with a power exponent for example: VSS·TSS-1.

Line 348, 349, 350, 353, 360. „….from 0.17 to 0.24% relative abundance at 42–48°C. ….”
The unit of measure and value should be separated by a space (0C). The unit of measure should be present with both range values.

Line 399, 402, 404, 424. Figure 8. „SRB/Methanogen”. Indication of the unit of measure "SRB/Methanogen” using a fractional dash is acceptable but is colloquial in meaning. However, exponential notation should be used.

Please use the notation of the quotient in units of measure using mathematical notation with a power exponent for example: SRB·Methanogen-1, SRB·M-1 .

Line 389, 392, 404, 424. „….M/TP ratio…..” Indication of the unit of measure "Methanogen/Total population” using a fractional dash is acceptable but is colloquial in meaning. However, exponential notation should be used.

Please use the notation of the quotient in units of measure using mathematical notation with a power exponent for example: M·TP-1.

Line 192, 214, 216, 222, 224, 229, 236, 238, 247, 263, 264, 274, 288, 289, 299, 300, 301, 306, 307, 308, 312, 313, 318, 319, 324, 326, 329, 349, 350, 351, 352, 362, 366, 367, 371, 369, 427, 431, 432. „….42°C…..” The unit of measure and value should be separated by a space.

Line 457. „….70.5 L/day to 28.6 L/day…..” Indication of the unit of measure "L/day” using a fractional dash is acceptable but is colloquial in meaning. However, exponential notation should be used.

Please use the notation of the quotient in units of measure using mathematical notation with a power exponent for example: L·day-1.

We should mark the „litre” with a small „l”.

Line 466, 467. „….31.24 g/l…..” Indication of the unit of measure "g/l” using a fractional dash is acceptable but is colloquial in meaning. However, exponential notation should be used.

Please use the notation of the quotient in units of measure using mathematical notation with a power exponent for example: g·l-1.

Line 482. „….42 to 45°C….” The unit of measure should be present with both range values. 

Line 458, 461, 462, 484, 490, 491. „…37–55°C…” The unit of measure should be present with both range values. 

Line 444, 445, 448, 449, 452, 455, 464, 482, 496, 497. „….35°C…..” The unit of measure and value should be separated by a space.

Line 525 – 526. „….(4.55 L/gCOD) to 152 ml (1.52 L/gCOD) then to 86.57 ml (0.94 L/gCOD)……” Indication of the unit of measure "L/gCOD” using a fractional dash is acceptable but is colloquial in meaning. However, exponential notation should be used.

Please use the notation of the quotient in units of measure using mathematical notation with a power exponent for example: L·g-1 COD.

The name of the parameter (COD) and its unit of measurement (g) should be separated by a space.

Line 527. „….42°C…..” The unit of measure and value should be separated by a space.

Response 2: Thank you very much for the noteworthy correction. We agree using mathematical notation with a power to express the unit of measurement. We also agree with the unit of measure and value should be separated by a space and present with both range values. We have incorporated the reviewer’s suggestions throughout the manuscript.

  • Point 3: The introduction of the genesis of the topic and the rationale is correct and does not dominate the content of the abstract. The goal and scope of the research are presented in a very laconic way. The research methodology is introduced through information about the realization of the main theses of the work. In general, the abstract is understandable for the reader. However, the abstract does not reflect the multi-threaded structure of the scientific work. Lack of direct reference to sewage sludge as a test substrate. Please take this into account.

Response 3: Thank you very much for the advise. We have emphasized the usage of sewage sludge as the microorganisms source through the sentence “This study investigated the effects of temperature shifts on CH4 gas production and microbial diversity during anaerobic digestion of anaerobic sewage sludge using a metagenomic approach”. We also added a line to encompasess the multi-threaded structure of the scientific work. However, all in all, we encountered several difficulties when it comes to restructure the abstract to the multi-threaded scientific work structure due to word counts limitation of 200 words.

  • Point 4:„COD” – Please explain the introduced abbreviations of terms, possibly as soon as they are introduced in the content of the article. A single explanation is sufficient in the execution of the explanation. For example:

COD – chemical oxygen demand.
Line 150. „chemical oxygen demand (COD) of 2000 mg/l was…” The explanation of the abbreviated name appears first on line 150. Authors may leave the explanation on line 150 or accelerate the explanation on line 19. A single explanation of the abbreviated term in the text of the article is sufficient.

Line 120-121, 127, 128, 129, 130. „…..inoculum to substrate ratio (I/S), and the I/S ratio…..”
Inoculum (I), substrates (S), please complete the abbreviations (symbols) in the text.

Line 172. „…DNA….” Please explain the introduced abbreviations of terms, possibly as soon as they are introduced in the content of the article. A single explanation is sufficient in the execution of the explanation.
For example: DNA: genetic code, Deoxyribonucleic acid?

Line 176. „….RNA…” Please explain the introduced abbreviations of terms, possibly as soon as they are introduced in the content of the article. A single explanation is sufficient in the execution of the explanation.
For example: RNA: genetic code, ribonucleic acid (RNA)

Line 178. „….into cDNA….” Please insert an explanation in the text, for example "Complementary DNA"

Line 272 „….DNA….”. Please ensure that the abbreviated name is explained beforehand.

Line 294. "ANOVA" please explain the abbreviation, for example (analysis of variance)

Line 294, 295. „P-value”, please explain the abbreviation, for example (probability value)

Line 130, 206, 234, 410, 474. „….VFAs…..” The abbreviation was described on 130 lines, correctly.

Line 500. „….VFA consumption….” The abbreviation was explained as VFAs meaning plural. Please explain the abbreviation again for a correct understanding of the article content.

Response 4: Thank you very much for the meticulous correction. We have incorporated reviewer’s suggestions throughout the manuscript.

  • Point 5: Keywordsare chosen correctly. Multiple keywords is not a mistake but requires an editor's opinion. Please take this into consideration.
    Response 5: Thank you for the suggestion. According to the IJERPH template, it is acceptable to put 5-10 keywords on the manuscript.

  • Point 6: The rationale for undertaking the research topic is logical and comprehensively documented. The order of citing literature sources in the chapter is correct. No critical comments on the scientific level of the chapter.
    Response 6: Thank you very much, we are grateful to hear that.

  • Point 7: The methodological assumptions and selection of research tools are correct. The methodological assumptions of the study are well considered and well documented. There is a lack of information on the moment of application and the importance of dividing the research into the part: with shifted temperature and controlled temperature. Please clearly define both research situations already in chapter 2.

Response 7: We would like to thank for reviewer’s positive feedback. We are aware with the lack of explanations about the research methodologies. Hence, we already put several lines to explains the controlled temperature research. Please kindly read it in Materials and Methods section, Experimental Procedures subsection.

In addition to the comment, we also apologize that we made two confusing statements by including shifted-down research method in the Material and Methods section in our previous manuscript. We did have shifted-down research, however due to several considerations, we decided to exclude the shifted-down data from the manuscript. Yet we missed to dispatch these two statements from the Materials and Methods section. We have deleted the statements now as can be seen in Materials and Methods section, Experimental Procedures subsection.

  • Point 8: Line 115, 116. „….FeCl3.6H2O….” I propose a clear mathematical notation with a dot, for example FeCl36H2O

Response 8: Thank you very much for the correction. We have applied it to all of the hydrated chemicals.

  • Point 9: Line 122. Table 1. The unit of measurement for pH is pH= –log10[a(H+)]

Response 9: Thank you very much for the information, we have inserted it to the table.

  • Point 10: Line 130-131. „….approximately 3 ml/1 ml…..” The notation with a slash as a symbol for division (ratio symbol) is rather unsuggestive and colloquial.

Response 10: We agree with this correction, we have restructured the sentence.

  • Point 11: The research findings are evidently multithematic in nature. There is a lack, in earlier chapters, of sufficient announcement of research themes. In its present form, the research findings are surprising due to the lack of preparation of an adequate research scope. Please take this into account. Presentation of results logical and correct.

Response 11: Thank you for the positive feedback. We are aware that this incongruity was probably caused by the misleading statements and some missing explanations in Material and Methods section. Hence, we have deleted several lines from previous manuscript and inserted new lines that sincerely will helps to emphasize the research.

Nevertheless, we respectfully disagree with the notion that this research was lack of an adequate research scope. In the introduction we have clearly proposed the aim of our study by saying “Notably, investigation of biogas production and genomic analysis from the anaerobic digestion process under shifted-up temperatures has not been performed till date. Therefore, there is an urgent need to expand our understanding of this field. Owing to the heat-resistant characteristics and possible benefits of AD in WWTP-scale applications, further research regarding thermotolerant microorganisms needs to be performed, especially to investigate their potential for biogas production. Therefore, the objectives of this study were to investigate the potential of biogas production (especially CH4) and to identify the most biogas-producing microorganisms among the cultures after being shifted to several temperature levels.” We think these sentences in introduction has carefully explained our focus on this research.

  • Point 12: CODadded. Please explain the abbreviation „added” in the bottom apostrophe.

Response 12: We realized this appostrophe may caused confusion, hence we have replaced it with the word “COD feed” as it already mentioned in the previous section. The words “feed” and “added” have the similar meaning of “the amount of COD of substrate that was added to the reactor for every feeding periods”. Thank you for pointing this out.

  • Point 13: Line 250-251. „…Figure 2. The production of CO2was spiking after the 3rd fed while the CH4 production was below 4 mL CH4 in 12 consecutive days. ….” The name of the figure is more a commentary than information about what characterises the figure. Please correct this.

Response 13: We agree with this, therefore we changed it to “The production of CO2 after the 3rd fed compared to CH4 production in 12 consecutive days.”

  • Point 14: Line 268. Please repeat the explanation of the abbreviated names in Table 2:

TSS – total suspended solids

VSS – volatile suspended solids

Response 14: Thank you very much for the meticulous correction. We have incorporated reviewer’s suggestions throughout the manuscript.

  • Point 15: Line 338-340. Figure 5. A better designation of the ordinate axis is "Bacterial count" instead of "Methanogenic bacteria". The axis designation should be unambiguous.

Response 15: We have revised it accordingly to improve the point on the figure.

  • Point 16: Line 341 -344. Figure 6. a), b) c). Lack of unit of measure for the temperature parameter for the „abscissa” axis (OX axis).

Line 416 Figure 7 a), b). Lack of unit of measure for the temperature parameter for the „abscissa” axis (OX axis).

Response 16: Thank you for pointing this out. We have inserted the unit of measurement to improve the point on the figure.

  • Point 17: Line 430-431. „…..from 2–0.75–430 0.59% and 6.52–4.96–3.81%.....” The range notation is ambiguous. Please use a more intuitive form of presenting the results.

Response 17: We agree with this point and have revised it in the manuscript. Please see it in page 16.

  • Point 18: Discussion of the results conducted correctly based on the discussion of own results and the current state of knowledge. Multiple attempts to explain the findings indicate the proficiency of the authors in the research topic.

Response 18: We would like to thank for reviewer’s positive feedback.

  • Point 19: 86 ml →94 L please verify dissonance of values.

Response 19: Thank you for noticing the dissonance of values, 0.94 l.g-1 COD is a specific methane yield that was calculated using methane volume production (86 ml), feeding flowrate, and COD concentration of each feed. This 0.94 was the correct number, we were mistakenly writing 0.86 l.g-1 COD in results and discussions in previous manuscript and has revised it accordingly in the current manuscript.

Reviewer 2 Report

This paper addresses a critical requirement for conducive anaerobic digestion (AD) and its effects on the production of biogas. The control of the operating temperature in the AD is crucial to ensure a stable and efficient process, and culminates in a good and steady production of biogas as a result of a good methanogenic activity. However, this study lacks originality and novelty. This should be stressed from the get-go, and further sustained in the text. However, despite the necessity to improve the quality of the figures, the results were well discussed and compared to previous studies that intersect with this manuscript. Please consider the following comments for the improvement of the overall quality of the paper.

Line 12: Please be careful with such statement, which is not completely true depending on the prevailing temperature during the summer from one region of the planet to another, as the operating temperature in anaerobic systems can also be maintained by a good insulation, which can prevent significant heat gain and/or loss from heating system. This is a much cheaper solution than using a cooling insulation system. 

Line 71 -78: There is an energy cost and revenue trade-off, when comparing which option is attractive, between operating an anaerobic digestion system under mesophilic and thermophilic temperature ranges. Several studies have suggested that, although the production of biogas is usually higher at thermophilic temperatures, the expenses related to the maintenance of such system, depending on the region, might impede the overall revenue/cost ratio from such systems. Furthermore, it is also reported that AD under a mesophilic temperature range is more stable than a thermophilic one. However, I agree with the authors that the production of biogas given a certain substrate concentration is definitely higher at a thermophilic temperature range.

Lines 154- 155. During the temperature shift-down process, why was the temperature reduced and then increased? Wasn't it more appropriate for the sake of evaluation of the effects of the decrease in temperature to lower it consistently at incremental ranges?

Line 206: What was the delta in the drop of the pH?

The quality of figures in the text should be improved, some of them look blurry and others lack an error margin when reporting results. Also, some figures should be more intuitive.

Figure 3: Did the biogas have the same composition in the three cases? Could be interesting to monitor the concentration of each element with the variation of the temperature throughout the study. 

Figure 8: why are the line plots of the methane production discontinued?

Author Response

Response to Reviewer 2 Comments

Dear reviewer, we would like to thank you for the noteworthy feedbacks. We appreciate the time and effort that you have dedicated to providing valuable feedback on my manuscript. We have been able to incorporate changes to reflect most of the suggestions provided by the reviewer. We have highlighted the changes in the manuscript using Track Changes function according to Editor’s suggestion.

Here is a point-by-point response to the reviewer’s comments and concerns. The letters written on black ink are the concerns, comments, and questions. While the letters written on red ink are the responses from us as the authors.

  • Point 1: This paper addresses a critical requirement for conducive anaerobic digestion (AD) and its effects on the production of biogas. The control of the operating temperature in the AD is crucial to ensure a stable and efficient process, and culminates in a good and steady production of biogas as a result of a good methanogenic activity. However, this study lacks originality and novelty. This should be stressed from the get-go, and further sustained in the text.

Response 1: We thank the reviewer for the feedback. However, we do not entirely agree with the reviewer’s argument that stated this study lacks of originality and novelty. We acknowledge that there have been a number of previous researches that discussed about the influence of temperature on AD performance, nevertheless most of the researches were conducted at a stabilized temperature conditions. Only a few explored the effect of stepwise temperature shifts on AD performance and microbial communities extensively. Westerholm et al. [1] and Beale et al. [2] are, to the best of our knowledge, the most recent researches that profoundly studied the effects of shifted-up temperature on biogas production and microbial communities. Nevertheless, they emphasized their research on the one-point temperature shifts manner, instead of profound study on the stepwise temperature shifts. Previous researches also mostly concentrated on examining the impacts of a big leap from mesophilic to thermophilic on biogas production, while our study was the first to explore the biogas production and microbial communities on 42 °C – 48 °C AD zone, on a stepwise shifts manner. This zone was inspired by findings from Westerholm et al. [1] that discovered immense perturbations around 40 – 44 °C suggesting further studies on this area. Yet the researchers did not specify the abundance of methanogens and left an unanswered question of “in which temperature (around 40 – 44 °C) did the inhibition happened?”. In our research, we found that the AD still performed well at 42 °C while disturbances occurred at 45 °C. Study from Pap et al. [3] and Moestedt et al. [4] also discovered similar inhibition around 40 – 44 °C however they used stable temperature conditions and did not particularly observed the microbial communities. Hence, we think this study has a considerable novelty and a valuable contribution to the field of AD.

  • Point 2: Line 12: Please be careful with such statement, which is not completely true depending on the prevailing temperature during the summer from one region of the planet to another, as the operating temperature in anaerobic systems can also be maintained by a good insulation, which can prevent significant heat gain and/or loss from heating system. This is a much cheaper solution than using a cooling insulation system.

Response 2: We agree with the reviewer’s assessment. We are aware that the statement, pointed out by the reviewer, was too strong and somehow irrelevant to other regions. Our statement was made according to previous studies and some cases in our region WWTP, meanwhile there are several countries, or regions, that did not experience the prevailing temperature during a particular season and have a better and cheaper solution towards temperature instability of AD process. Therefore, we have revised the statement according to the reviewer’s comment to avoid irrelevance to other cases that contradicts our statement.

  • Point 3: Line 71 -78: There is an energy cost and revenue trade-off, when comparing which option is attractive, between operating an anaerobic digestion system under mesophilic and thermophilic temperature ranges. Several studies have suggested that, although the production of biogas is usually higher at thermophilic temperatures, the expenses related to the maintenance of such system, depending on the region, might impede the overall revenue/cost ratio from such systems. Furthermore, it is also reported that AD under a mesophilic temperature range is more stable than a thermophilic one. However, I agree with the authors that the production of biogas given a certain substrate concentration is definitely higher at a thermophilic temperature range.

Response 3: We would like to thank the reviewer for the explanation regarding technical and non-technical considerations between mesophilic and thermophilic operated AD and for the supporting ideas regarding to the better performance at higher temperature range. It would be interesting to explore this aspect any further. Nevertheless, in the case of this study, our research was firmly concentrated on observing the AD performance in term of biogas production during shifted-up temperature conditions and the effect of this temperature shifts on microbial communities (genomic study).

  • Point 4: Lines 154- 155. During the temperature shift-down process, why was the temperature reduced and then increased? Wasn't it more appropriate for the sake of evaluation of the effects of the decrease in temperature to lower it consistently at incremental ranges?

Response 4: In this research we focused on the upshifted temperature effects on biogas production and microbial communities, hence the temperature was adjusted increasing only. We apologize that we made two confusing statements by including shifted-down research method in the Material and Methods section in our previous manuscript. We slighted to remove these two statements from the Materials and Methods section. In the present manuscript, we have deleted the statements as can be seen in Materials and Methods section, Experimental Procedures subsection. In addition, we wrote some details of controlled temperature research in Materials and Methods section that we conducted as comparison to shifted-up research.

We appreciate the reviewer’s view that mentioned evaluating a decrease in temperature may be more appropriate. However, we think that evaluating the effects of shifting up the temperature also have an equal importance because several previous researches reported higher biogas yield on a higher temperature levels. According to previous studies, temperature accelerates metabolic rates and biochemical processes, the higher the temperatures have been demonstrated to enable higher degradation rates and higher biogas yields from a wide variety of substrates [1,5–7]. This statement was our initial consideration of starting the shifted-up temperature along with the curiousity of finding thermotolerant methanogens (methanogenic archaea that can survive well in multiple thermal changes).

  • Point 5: Line 206: What was the delta in the drop of the pH?

Response 5: The statement that we wrote in line 206 was cited from the research conducted by El-Gendy et al. [8]. We used reference from literature to address the potential pH influence on CH4 production. We did not examine the daily pH change due to the limitation on sample available on the reactor. Moreover, we were trying to prevent frequent contact between sewage sludge sample on the AD reactor to any substances or materials from the outside of the reactor to avoid contamination and perturbations.

  • Point 6: The quality of figures in the text should be improved, some of them look blurry and others lack an error margin when reporting results. Also, some figures should be more intuitive.

Response 6: Thank you very much for the advice on improving the quality of the figures. We have accordingly modified several graphs using Origin 2020 in order to enhance the pixels quality. Please kindly check on our revised manuscript.

  • Point 7: Figure 3: Did the biogas have the same composition in the three cases? Could be interesting to monitor the concentration of each element with the variation of the temperature throughout the study. 

Response 7: There was a decrease in methane content during shifted-up temperature. We agree that it could be more interesting to show the difference of biogas compositions after temperature shifts. Hence, we have added several figures for this purpose, please kindly check in our revised manuscript.

  • Point 8: Figure 8: why are the line plots of the methane production discontinued?

Response 8: We tried to emphasize the different treatment conditions of each research reactor (shifted-up temperature conditions and controlled temperature conditions) using a discontinued line plot. Controlled temperature was conducted to show the biogas production and microbial communities when the reactors were operated in stable temperature conditions.

REFERENCE

  1. Westerholm, M.; Isaksson, S.; Karlsson Lindsjö, O.; Schnürer, A. Microbial Community Adaptability to Altered Temperature Conditions Determines the Potential for Process Optimisation in Biogas Production. Appl. Energy 2018, doi:10.1016/j.apenergy.2018.06.045.
  2. Beale, D.J.; Karpe, A. V.; McLeod, J.D.; Gondalia, S. V.; Muster, T.H.; Othman, M.Z.; Palombo, E.A.; Joshi, D. An “omics” Approach towards the Characterisation of Laboratory Scale Anaerobic Digesters Treating Municipal Sewage Sludge. Water Res. 2016, doi:10.1016/j.watres.2015.10.029.
  3. Pap, B.; Györkei, Á.; Boboescu, I.Z.; Nagy, I.K.; Bíró, T.; Kondorosi, É.; Maróti, G. Temperature-Dependent Transformation of Biogas-Producing Microbial Communities Points to the Increased Importance of Hydrogenotrophic Methanogenesis under Thermophilic Operation. Bioresour. Technol. 2015, doi:10.1016/j.biortech.2014.11.021.
  4. Moestedt, J.; Rönnberg, J.; Nordell, E. The Effect of Different Mesophilic Temperatures during Anaerobic Digestion of Sludge on the Overall Performance of a WWTP in Sweden. Water Sci. Technol. 2017, doi:10.2166/wst.2017.367.
  5. Yu, D.; Kurola, J.M.; Lähde, K.; Kymäläinen, M.; Sinkkonen, A.; Romantschuk, M. Biogas Production and Methanogenic Archaeal Community in Mesophilic and Thermophilic Anaerobic Co-Digestion Processes. J. Environ. Manage. 2014, doi:10.1016/j.jenvman.2014.04.025.
  6. Gebreeyessus, G.D.; Jenicek, P. Thermophilic versus Mesophilic Anaerobic Digestion of Sewage Sludge: A Comparative Review. Bioengineering 2016.
  7. Moset, V.; Poulsen, M.; Wahid, R.; Højberg, O.; Møller, H.B. Mesophilic versus Thermophilic Anaerobic Digestion of Cattle Manure: Methane Productivity and Microbial Ecology. Microb. Biotechnol. 2015, doi:10.1111/1751-7915.12271.
  8. El-Gendy, N.S.; Nassar, H.N. Biosynthesized Magnetite Nanoparticles as an Environmental Opulence and Sustainable Wastewater Treatment. Sci. Total Environ. 2021.